# OpenReview forum: "When Unlearning Backfires: Partial Unlearning Increases PII Regurgitation and enables data extraction in Meta’s Llama 3.2 1B"
_ICLR.cc/2026/Conference — Submitted to ICLR 2026_

### Official Review · Reviewer_Asrk · 2025-10-29

**Soundness:** 3
**Presentation:** 2
**Contribution:** 2
**Rating:** 4
**Confidence:** 4

**Summary:**

The paper investigates the unintended safety risks of partial unlearning in LLMs with a case study on Llama-3.2-1B-Instruct in the Who's Harry Potter [1] setting. By applying unlearning on only the Harry Potter novels, a subset of full Harry Potter knowledge source, the study simulates real-world unlearning where full training data is inaccessible. While the method effectively erases most Harry Potter references, it unexpectedly increases the model's tendency to produce unexpected outputs, such as PII, which the authors hypothesize results from the model regurgitating memorized training data.

The paper highlights that partial unlearning can paradoxically undermine safety and calls for further research into the unintended consequences of unlearning methods.

[1] "Who's Harry Potter? Approximate Unlearning in LLMs", Eldan & Russinovich, 2023.

**Strengths:**

1. **Clear methodology:** The paper follows a well-established unlearning setting, and provides detailed experimental parameters.

2. **Good conceptual framing:** The study highlights partial unlearning as a realistic setting, and attempts to connect the practical unlearning limitations with concrete safety risks such as PII regurgitation.

**Weaknesses:**

1. **Questionable conclusion:** The link between partial unlearning and increased PII leakage is not convincingly demonstrated. Section 4.2.1 appears to show two types of leakage: (1). public topic-related content (the Harry Potter fan blog example), and (2). genuinely unrelated PII (the university student example), which constitutes the true safety concern. I don't see evidence directly connecting (2). and partial unlearning from the study, and it remains plausible that similar leakage could occur even with comprehensive unlearning to a full forget set.
   - Questions:
      1. Can you provide additional evidence or analysis supporting that PII leakage specifically results from partial unlearning? Have you tested whether similar leakage occurs under complete unlearning?
      2. Can you quantify or categorize the leaked content to distinguish between benign and genuinely harmful PII leakage?


2. **Limited generality:** the experiments are limited to a single model, a single unlearning domain, and a single unlearning method, which constrains the generalizability of the conclusions. While the compute constraints are understandable, other methods such as RMU [1] and SatImp [2] can be implemented within the stated single A100 budget.
   - Questions:
      1. Can you clarify why Llama-3.2-1B-Instruct was chosen over other models, and whether you expect the same behaviors at larger scales?
      2. Is it feasible to conduct even a lightweight ablation on a different domain/unlearning method to verify whether the phenomenon generalizes beyond the speific setup in the paper?


Some other weaknesses that are less important:


3. **Insufficient analysis:** The decision to withhold red teaming prompts and model weights is defensible for safety, but it limits reproducibility and may hinder a full understanding or verification of the proposed findings.
   - Questions:
      1. Can you provide more qualitative and quantitative evaluations for section 4.2.1 and 4.2.2?


In conclusion, I think the empirical scope and the provided anlysis of the paper do not yet support the strength of its main conclusion. Strengthening the experimental breadth and clarifying the relation between harmful PII leakage and partial unlearning would substantially improve the paper's impact and credibility.

[1]. "The WMDP Benchmark: Measuring and Reducing Malicious Use With Unlearning", Li, et. al., 2025

[2]. "Exploring Criteria of Loss Reweighting to Enhance LLM Unlearning", Yang, et. al., 2025

**Questions:**

Please see the questions in weaknesses.

---

### Official Review · Reviewer_6EFc · 2025-10-31

**Soundness:** 1
**Presentation:** 1
**Contribution:** 2
**Rating:** 2
**Confidence:** 4

**Summary:**

In this paper, LLM unlearning is performed to make a model forget about some domain of knowledge. The paper shows that when the model is then queried about this knowledge, instead of outputting its original knowledge, it tends to output information about people who are related to the original topic, which could be interpreted as a form of PII leakage.

**Strengths:**

The premise of the paper is interesting. The question of what unintended consequences arise from unlearning is worth exploring.

**Weaknesses:**

The experiments in this paper are highly limited, only exploring a single model (Llama 3.2 1B), a single unlearning method that is not very standard, and a single unlearning target (Harry Potter knowledge). With such a narrow scope, we do not know if these results generalize beyond this single model + method + dataset setup. In fact, it seems clear that this PII leakage happens in Harry Potter because of the amount of fanfiction on the web, and the fact that fanfiction tends to mention PII such as information about the fanfiction author. However, in other domains this would be less likely. So the principle is not that unlearning leads to PII leakage, but just that unlearning leads to the model outputting content from the most related data that was not unlearned.

The only datasets used for evaluation are one dataset with 10 examples and a second dataset with 19 examples. Both are extremely small for any study.

The overall writing of this paper is quite non-standard and would benefit from editing by an experienced author. It seems to also be a consequence of the fact that there are very few empirical results, so instead there is a great deal of exposition about background and analysis of individual model outputs.

Finally, there are several minor formatting issues that should be addressed:
* Line 032: The citation for Lucki et al. is not in the right place relative to the end of the sentence.
* Line 079: Use $x$ in LaTeX instead of just plain x, when referring to a mathematical quantity named $x$.
* Line 087: Use ` instead of ' for a forward single quote
* Line 097: Missing citation
* Line 097: Use \citet when the citation is part of the sentence, e.g., "\citet{shi2024} incorporated the unlearned model from \citet{eldan2023}..."
* Line 215: These should be $v_{\text{baseline}}$ and $v_{\text{reinforced}}$ (similar to how they are in equation 1)
* Line 291: Capitalize proper names

While having a few such issues is not a problem, the consistent presence of these issues suggests a lack of care by the authors.

**Questions:**

Is 8% PII leakage high enough to be considered "often" (Line 335)? I ask especially because if this information does come from public fanfiction posts, these are things that the authors intentionally posted online. This doesn't seem as concerning as PII that is leaked by a third party and then memorized by an LLM.

Would these results hold for other models, unlearning methods, and domains? A broader investigation of this could lead to a good conference submission. As is, the narrow scope of the work makes it at best suitable for a workshop focused on unlearning.

---

### Official Review · Reviewer_mTia · 2025-10-31

**Soundness:** 1
**Presentation:** 1
**Contribution:** 1
**Rating:** 0
**Confidence:** 4

**Summary:**

The paper studies partial unlearning in LLMs, removing only a subset of the target data (the seven Harry Potter novels) from Meta’s LLaMA 3.2 1B using the Eldan & Russinovich (2023) method. The authors report that such incomplete unlearning surprisingly increases training-data regurgitation, including personally identifiable information (PII) from unrelated sources.

**Strengths:**

* Unlearning on incomplete forget sets is realistic and highly relevant -- most practical LLM unlearning scenarios are partial.
* The observation that partial unlearning can increase memorization is surprising and thought-provoking. Examples of PII leakage underline real safety risks and link unlearning to privacy and trustworthiness debates.

**Weaknesses:**

* The paper is poorly structured and difficult to follow; results are anecdotal and not clearly quantified (the paper doesn't contain a single figure or table). The methodological description is verbose but lacks conceptual clarity, most key design decisions are not justified.
* The central claim -- that unlearning the exact Harry Potter books leads to model producing Harry Potter content unprompted (Section 4.1.1) -- is counter-intuitive and potentially symptomatic of implementation or evaluation flaws. The intuition behind the result, and why it is so different from Eldan & Russinovich (2023) is unclear. It could potentially hint is issues with experiment design or implementation.
* The experimental setup is narrow, relying on a single text corpus (the Harry Potter novels) and a single small model (LLaMA 3.2 1B). This limits generality and makes it unclear whether the observed behavior would hold for other domains, scales, or architectures.

Overall, while the topic is timely, the paper lacks the methodological rigor, quantitative depth, and clarity of exposition required for a strong empirical contribution.

**Questions:**

N/A

**Details Of Ethics Concerns:**

Use of copyrighted material (the full novels) without sharing data or model checkpoints raises ethical and reproducibility concerns; the decision to withhold models and red-teaming prompts further limits verifiability.

---

### Official Review · Reviewer_KMTZ · 2025-10-31

**Soundness:** 2
**Presentation:** 2
**Contribution:** 4
**Rating:** 4
**Confidence:** 4

**Summary:**

This work studies the effect of targeted removal of a “core” forget set for unlearning that is not representative of the full intended unlearned topic. In particular, they study how unlearning a subset of the unlearning topic (in this case, the core Harry Potter book series text) impacts regurgitation of related information (such as blog posts about Harry Potter). They do this by reinforcing the Harry Potter-related information in the LLM and subsequently removing the added Harry Potter knowledge by altering the model’s logit distribution and then finetuning on generic alternatives. They find that partial unlearning results in increased prevalence of related information that contains personally identifiable information, such as the names, affiliations, and websites of blog post authors.

**Strengths:**

- The fundamental motivation of this work, understanding the effects of partial unlearning in realistic setups, is very interesting and under-explored in current unlearning evaluations.
- The results are novel and would be a contribution to the general unlearning community.

**Weaknesses:**

- The authors test a single model on a single method for a single dataset. Why do they not compare multiple unlearning methods? What motivates the use of the one that the authors propose? How does it differ from Eldan and Russinovich’s?
- The authors provide no intuition or discussion for why this behavior comes about. Is this a result of unlearning only the core text of Harry Potter without unlearning any auxiliary data? If the authors had partially unlearned other related sources (blog posts, fanfiction, and Wikipedia entries), would this behavior still exist?
- Is the PII extractable (via red-teaming) even for the base model and the reinforced model before unlearning?

Generally, while I find this work well-motivated and very interesting, the experiments are not thorough or generalizable. It is difficult to understand if the observed behavior is a result of partial unlearning, the specific data that the authors unlearned, the specific unlearning method that the authors used, or even simply the setup of their red teaming. Without comparing each of the model versions (original, reinforced, unlearned) across regular evaluations, Harry Potter-centered evaluations, and red-team attacks, it is hard to draw conclusions. Furthermore, very little justification is provided for the unlearning method that the authors use, and no comparison is made with any other unlearning methods or different core forget sets.

I believe this paper would strongly benefit from both a more thorough empirical analysis of the observed behavior and a discussion of why this behavior may exist.

**Questions:**

- Missing citation line 97
- Can the authors make the distinction between their unlearning method and Eldan and Russinovich’s more clear?

---

### Meta-Review · Area_Chair_ZXqm · 2026-01-02

**Summary:**

Reviewers found the topic timely and the observed phenomenon potentially interesting. However, there is broad agreement that the paper’s empirical scope is extremely limited, relying on a single small model, a single unlearning method, and a single domain. The causal link between partial unlearning and increased PII regurgitation is therefore not convincingly established, and alternative explanations (data/domain effects, method-specific behavior, or red-teaming artifacts) cannot be ruled out. In addition, concerns were raised about weak quantitative evaluation, unclear methodology, and ethical issues related to copyrighted data and PII extraction. Since these core concerns were not addressed due to the absence of a rebuttal, they remain outstanding and directly inform the rejection decision.

**Reviewer Concerns:**

The authors did not provide a rebuttal. As a result, none of the substantive reviewer concerns were addressed.

**Reviewer Scores:**

The authors did not provide a rebuttal. As a result, none of the scores may be changed.

---

### Decision · Program_Chairs · 2026-01-26

Reject